# Early Assessment of Voice Problems in Post-Thyroidectomy Syndrome Using Cepstral Analysis

**DOI:** 10.3390/diagnostics14010111

**Published:** 2024-01-04

**Authors:** Yeso Choi, Bo Ram Keum, Ju Eun Kim, Joong Seob Lee, Seok Min Hong, IL-Seok Park, Heejin Kim

**Affiliations:** 1Department of Otorhinolaryngology-Head and Neck Surgery, Dongtan Sacred Heart Hospital, Hallym University of Medicine, Hwaseong 18450, Republic of Korea; 201080@hallym.or.kr (Y.C.); monet02@hallym.or.kr (B.R.K.); llbeaura@hallym.or.kr (J.E.K.); thecell@hallym.or.kr (S.M.H.); ispark@hallym.or.kr (I.-S.P.); 2Department of Otorhinolaryngology-Head and Neck Surgery, Hallym Sacred Heart Hospital, Hallym University of Medicine, Anyang-si 14068, Republic of Korea; apniosio@naver.com

**Keywords:** post-thyroidectomy syndrome, vocal fatigue, cepstral analysis

## Abstract

Post-thyroidectomy syndrome (PTS), characterized by voice issues after thyroidectomy without recurrent laryngeal nerve injury, was investigated in this study. The Voice Fatigue Index (VFI) and cepstral analysis were employed for subjective and objective voice evaluation. Retrospective analysis involved 96 patients (37 males, 59 females) who underwent thyroidectomy without nerve injury from April 2018 to June 2022. Assessments pre- and post-thyroidectomy included the Voice Handicap Index (VHI) and VFI, along with auditory perceptual, acoustic (including cepstral), aerodynamic, and glottal vibration analyses. In females, although the GRBAS scale showed no significant change, both VHI and VFI increased post-thyroidectomy. Significant correlations were observed between the VHI and VFI in females. Acoustic analysis indicated a decrease in the cepstral peak prominence (CPP) of vowels (/a/) and sentences in females, with significant correlations between changes in the CPP/a/ and VHI/VFI. The maximum fundamental frequency (F_0_max) exhibited a significant decrease, correlating with the VHI and VFI changes. The VFI demonstrated effectiveness in subjective PTS voice evaluation, comparable to the VHI. The present study highlights the potential of cepstral analysis as an index reflecting subjective voice discomfort, suggesting its promise for a comprehensive PTS voice evaluation.

## 1. Introduction

According to statistics from the National Cancer Information Center in 2020 [1], thyroid cancer ranks first in prevalence, comprising 11.8% of all cancers. On the other hand, given the 100% 5-year survival rate of thyroid cancer, there is an increasing demand among patients for a better quality of life after thyroidectomy. The utilization of nerve monitoring during surgery and advancements in surgical equipment designed to minimize heat damage during thyroidectomy have contributed to a reduction in the rate of recurrent laryngeal nerve (RLN) damage. Nevertheless, many patients continue to experience voice discomfort even in the absence of damage to the RLN [2]. Previous studies have reported subjective negative changes in patients after thyroidectomy ranging from 29% to as much as 87% [3,4,5,6]. Post-thyroidectomy syndrome (PTS) is characterized by various symptoms, including swallowing issues, voice disorders, such as vocal fatigue, sensory abnormalities, changes in voice, and an inability to produce high-pitched sounds. Importantly, these symptoms occur without evidence of damage to the RLN following thyroidectomy [7]. The possible causes have been reported to be related to damage caused by tracheal intubation during surgery, damage to the external branch of the superior laryngeal nerve (EBSLN), laryngotracheal fixation, impaired blood circulation to the larynx, and delayed laryngeal lymph edema [7,8].

Several questionnaires have been developed and employed to measure patients’ subjective discomfort, aiming to analyze the voice problems arising after thyroid surgery. Representative examples include the Voice Handicap Index (VHI) [9], Voice-Related Quality of Life (V-RQOL) [10], Voice Fatigue Index (VFI) [11], and the Thyroidectomy-related Voice Questionnaire (TVQ) [12]. The VHI is the most widely used index in the study of patients with voice problems, such as vocal cord polyps, cysts, and nodules, extensively employed in previous studies to identify voice abnormalities. However, patients with PTS do not exhibit abnormalities in vocal cord movement, raising concerns about relying solely on the negative VHI, especially when assessing vocal fatigue after surgery. On the other hand, the GRBAS scale [13] is employed for auditory perceptual evaluation of voice, while changes in objective voice indicators can be verified through acoustical analysis using programs like the Multi-Dimensional Voice Program (Model 5105, KayPENTAX, Montvale, NY, USA, MDVP). The MDVP is commonly used for voice measurements, assessing time-based parameters such as jitter and shimmer, as well as frequency-based parameters like the noise-to-harmonic ratio (NHR). However, some limitations have been suggested, particularly regarding the reliability in connected speech, sustained vowels, and across a continuum of dysphonic severity [14]. To overcome the limitations of time-based analysis, cepstral acoustic analysis, assessed by the Analysis of Dysphonia in Speech and Voice (ADSV, Model 5109, KayPENTAX) and Praat (Institute of Phonetic Sciences, University of Amsterdam, Amsterdam, The Netherlands) programs, has evolved and been employed in various types of voice disorders. Cepstral analysis has advantages, in that it can estimate aperiodicity or additive noise without the need for the identification of cycle boundaries [15]. Therefore, measuring the cepstral peak prominence (CPP) using cepstral analysis can help identify dysphonia severity in sustained vowel productions and connected speech. Cepstral analysis is widely used for various voice disorders; however, there are only a few studies on the cepstral analysis of voice problems in patients with PTS.

In this study, our objective is to evaluate the utility of conventional subjective assessments, such as the VHI and VFI. Additionally, we conducted various objective tests, including the MDVP and cepstral analysis, to assess the negative indicators that accurately reflect subjective discomfort in patients’ voices.

## 2. Materials and Methods

### 2.1. Patients

This retrospective study analyzed the medical records of patients aged 18 years or older who underwent thyroidectomy at our hospital’s otolaryngology department from April 2018 to June 2022. The study received approval from the Institutional Review Board (IRB) of our hospital (IRB no. 2018-08-012-006). Exclusion criteria comprised patients with abnormal findings of vocal cords in preoperative laryngoscopy, abnormal vocal cord movement detected through intraoperative nerve monitoring during surgery, or abnormal vocal cord movement observed after surgery. Additionally, the study excluded patients with a history of neck infections or inflammation, those who had undergone neck surgery or radiation therapy, individuals with respiratory diseases, central nervous system diseases, or lingering aftereffects of previous neurological diseases.

In total, 96 patients were enrolled, with 37 males and 59 females. The mean age of males was 50.6 ± 10.9 years, while the mean age of females was 47.0 ± 9.9 years, showing no statistically significant difference (Table 1). Specifically, 24 males underwent thyroid lobectomy and 13 underwent total thyroidectomy. For females, 33 underwent thyroid lobectomy and 26 underwent total thyroidectomy. All patients underwent central neck dissection with thyroidectomy, and additional lateral neck dissection was performed in 5 males and 2 females. The thyroidectomies were performed by two experienced otolaryngologists.

Nerve monitoring during surgery involved placing two pairs of electrodes in the Medtronic nerve integrity monitor (electromyography, EMG) tracheal tube in contact with the vocal cords. Utilizing the Nerve Integrity Monitor (NIM) response 3.0 (Medtronic, Jacksonville, FL, USA), electrical stimulation was applied to the nerve to assess the EMG waveform and check for any potential nerve damage. Postoperatively, all patients underwent laryngoscopy to confirm normal laryngeal movement.

### 2.2. Voice Analysis

Both subjective and objective voice analyses were conducted before thyroidectomy and 1 month after thyroidectomy, and all of them were assessed by two experienced speech pathologists. Subjective voice analysis was conducted using the Korean version of the Voice Handicap Index (K-VHI) and the Voice Fatigue Index (K-VFI). The K-VHI comprises 30 questions to evaluate the patient’s voice discomfort, while the K-VFI consists of 19 questions to assess voice fatigue [9,11].

Auditory perceptual voice analysis was conducted using the GRBAS scales, which rated the grade (G), roughness (R), breathiness (B), asthenia (A), and strain (S) on a scale from 0 to 3 [13]. A Computerized Speech Lab (Model 4150B; KayPENTAX, Lincoln Park, NJ, USA; CSL), Real-time Electroglottography (Model 6103; KayPENTAX, Lincoln Park, NJ, USA; EGG), and a unidirectional dynamic microphone (SM48; SHURE, Niles, IL, USA) were used to record the voice in a room designed to block external noises. The microphone was positioned at a constant distance of 5 cm and stably mounted using a T-shaped microphone stand. The vowel /a/ was extended for more than 4 s, and a sentence containing Korean words was selected for reading. Acoustical analysis was performed using the Multi-Dimensional Voice Program (MDVP, Model 5105; KayPENTAX, Lincoln Park, NJ, USA) developed by the CSL. The analysis calculated parameters including fundamental frequency (F_0_), jitter, shimmer, the noise-to-harmonic ratio (NHR), and the Soft Phonation Index (SPI). Cepstral analysis was conducted using the Analysis of Dysphonia in Speech and Voice (ADSV, Model 5109; KayPENTAX, Lincoln Park, NJ, USA), which can measure the cepstral peak prominence (CPP) [16] and low-to-high spectral ratio (L/H ratio). Additionally, the vocal range profile (VRP) from the CSL was utilized to measure the minimum fundamental frequency (F_0_min) to the maximum fundamental frequency (F_0_max) to assess changes in the vocal range.

For the aerodynamic analysis, the maximal phonation time (MPT) was measured using a Phonological Aerodynamic System (PAS 6600; KayPENTAX, Montvale, NJ, USA). To observe vocal cord movement, a real-time EGG hardware was employed, measuring the closed quotient (CQ).

### 2.3. Statistical Analysis

IBM SPSS Statistics software (version 27.0; IBM Corp., Armonk, NY, USA) was utilized. Student’s *t*-test and the Chi-square test were employed for the analysis of patient demographic data. A paired *t*-test was conducted to analyze changes in subjective discomfort and negative index values before and after thyroidectomy. Additionally, a Pearson correlation analysis was applied to identify negative indicators correlating with subjective discomfort. The statistical significance level was set at *p* < 0.05 for all analyses.

## 3. Results

After thyroidectomy, both the K-VHI and the K-VFI exhibited statistically significant increases from 3.19 ± 7.5 and 4.03 ± 8.19 before surgery to 9.91 ± 16.3 and 9.17 ± 14.1 after surgery, respectively (*p* < 0.001 for both). When stratified by gender, in men, the K-VHI increased from 2.84 ± 6.8 before surgery to 6.43 ± 11.0 after surgery, and the K-VFI increased from 3.00 ± 6.7 before surgery to 4.54 ± 7.7 after surgery, though without reaching statistical significance. Conversely, in women, the K-VHI increased from 3.41 ± 8.0 before surgery to 12.08 ± 18.4 after surgery, and the K-VFI increased from 4.68 ± 8.8 before surgery to 12.07 ± 16.4 after surgery (*p* = 0.001 and *p* < 0.001, respectively, as shown in Figure 1).

After surgery, the G (grade) component of the GRBAS Scale, used for auditory perception evaluation, showed a statistically significant decrease in men, dropping from 0.19 ± 0.4 before surgery to 0.07 ± 0.21 after surgery. In contrast, in women, the G score rose from 0.17 ± 0.3 before surgery to 0.19 ± 0.4 after surgery, but this change was not statistically significant. No other statistically significant differences were observed in both men and women across the other scales, except for the G scale (see Table 2).

No statistically significant differences were observed in the F_0_, jitter, NHR, and SPI before and after surgery in both males and females, as determined through acoustic analysis using the MDVP. However, shimmer exhibited a significant decrease from 2.82 ± 1.3% before surgery to 2.29 ± 0.9% after surgery in men and from 2.65 ± 1.1% before surgery to 2.04 ± 0.7% after surgery in women (*p* = 0.002, *p* = 0.024; Table 3).

In the cepstral analysis, the CPP of the male /a/ vowel (CPP/a/) decreased, while the CPP of the sentence (CPP/sen/) increased (*p* = 0.903, *p* = 0.198). For the females, both CPP/a/ and CPP/sen/ decreased, but the differences were not statistically significant (*p* = 0.124, *p* = 0.851). In both males and females, the L/H ratio increased for sentences and the /a/ vowel, but only in females did the L/H ratio of sentences show a statistically significant increase, rising from 34.84 ± 3.6 before surgery to 36.20 ± 3.8 after surgery (*p* < 0.001, Table 3).

The minimum fundamental frequency (F_0min_) in males, measured using the VRP, increased from 86.83 ± 16.0 Hz before surgery to 88.03 ± 15.0 Hz after surgery. The maximum fundamental frequency (F_0max_) decreased from 262.75 ± 52.2 Hz before surgery to 255.42 ± 57.1 Hz after surgery, but showed no statistically significant difference (*p* = 0.566; *p* = 0.364). For females, the F_0_min decreased from 143.86 ± 19.6 Hz before surgery to 140.16 ± 20.8 Hz after surgery, but did not show a statistically significant difference (*p* = 0.111). However, the F_0max_ decreased significantly from 441.96 ± 88.0 Hz before surgery to 405.65 ± 100.0 Hz after surgery (*p* = 0.003, Table 3).

In the aerodynamic test, the MPT decreased from 17.26 ± 5.9 s before surgery to 16.01 ± 6.2 s after surgery in men, and for women, it also decreased from 15.29 ± 4.9 s to 14.26 ± 4.1 s; however, these reductions were not statistically significant (*p* = 0.209, *p* = 0.084). The closed quotient (CQ), measured by the EGG, did not show any significant difference in men. In women, it significantly decreased from 47.03 ± 3.0% before surgery to 45.19 ± 4.3% after surgery (*p* = 0.003, Table 3).

The correlation coefficient between the change in VHI (ΔVHI) and the change in VFI (ΔVFI), used to assess subjective voice discomfort, was 0.887, and this correlation was statistically significant (*p* < 0.001, Table 4).

In addition, the ΔVHI showed a statistically significant negative correlation with the Δ shimmer (correlation coefficient (*r*) of −0.221), ΔCPP/a/ (*r* = −0.315), ΔF_0_max (*r* = −0.393), and ΔCQ ((*r* = −0.520) (*p* = 0.030, *p* = 0.003, *p* < 0.001, *p* < 0.001, Table 5). Similarly, the ΔVFI was statistically significant with the ΔCPP/a/ (*r* = −0.325), ΔF_0_max (*r* = −0.452), and ΔCQ (*r* = −0.418) (*p* = 0.002, *p* < 0.001, *p* < 0.001, Table 5).

## 4. Discussion

Voice impairment without damage to the RLN following thyroidectomy has been observed in numerous studies, referred to as PTS or post-thyroidectomy voice disorder (PTVD). In a web-based survey involving 4426 patients who underwent thyroidectomy, 51.1% (1693 patients) reported various subjective voice problems [4]. In our study, 48.9% (47 out of 96 patients) exhibited an increase in the K-VHI and 47.9% (46 out of 96 patients) showed an increase in the K-VFI. These findings highlight that a substantial number of patients experience subjective voice impairment shortly after thyroidectomy, even in the absence of damage to the RLN.

The VHI is a widely used test that formulates questions in the domains of physical (P), functional (F), and emotional aspects (E), assessing the perception and severity of voice disorders [9]. It serves as a primary tool for subjective voice evaluation globally, and is also valuable for assessing patients after thyroidectomy. In a study by Ryu et al. [6], 130 (84%) out of 155 patients who underwent thyroidectomy reported an increase in the VHI by more than 10% after surgery. In a study by Papadakis et al. [17], involving the evaluation of 191 patients who underwent total thyroidectomy with a preserved RLN, an increase in the VHI was also observed from an average value of 4.2 before thyroidectomy to 8.5 after 1 week of total thyroidectomy. This increase subsequently decreased to 4.5 after 8 weeks.

As a result of a meta-analysis investigating patients without nerve damage after thyroidectomy, the most commonly reported symptom was vocal fatigue [5]. Vocal fatigue, recognized by the voice user, primarily reflects an increased perception of voice effort that intensifies with prolonged voice use and typically improves with rest [18]. The VFI, introduced in 2015 by Nanjundeswaran et al. [11], proves useful for evaluating vocal fatigue, categorizing it into three factors: factor 1 (tiredness and voice avoidance), factor 2 (physical discomfort during voice use), and factor 3 (improvement of symptoms after rest). Despite vocal fatigue being a pivotal symptom of PTS, there is a relative scarcity of studies investigating the relationship between the VFI and PTS. In a recent study assessing the correlation between the VHI and VFI, it was reported that a significant correlation exists between factor 1 and 2 of the VFI and VHI, while no correlation was observed between factor 3 of the VFI and VHI [19,20]. However, in our study, changes in all three factors of the VFI showed a significant correlation with changes in the VHI. Therefore, the VFI is anticipated to be a more delicate voice evaluation tool, especially in the context of thyroidectomy-related vocal fatigue. Most studies on voice impairment following thyroidectomy have demonstrated an elevation in the GRBAS scale postsurgery, and this increase is frequently proportional to the subjective evaluations provided by patients [2,17,21]. However, no significant change was observed in the GRBAS scale for patients in a study related to post-thyroidectomy [22]. Additionally, other studies on vocal fatigue [23,24] reported no difference in auditory perception evaluation after inducing vocal fatigue. In our study, we similarly observed no significant difference in the GRBAS scale after thyroidectomy. This not only suggests that the GRBAS scale may not be effective in measuring vocal fatigue, but also underscores its limitations as an examiner-centered evaluation. It can be inferred that the degree of voice discomfort is negligible in patients with PTS who lack abnormalities in the RLN, or that subjective discomfort outweighs objective voice changes.

While PTS presents a significant disparity in patients’ subjective voice discomfort, the corresponding objective voice parameter has not been conclusively identified. Myers EN et al. [8] reported no changes in the F_0_ and MPT in 54 patients who underwent thyroidectomy without RLN injury; however, there was a decrease in the speaking fundamental frequency (speaking F_0_). Sinagra DL et al. [3] reported a decrease in the F_0_ and an increase in shimmer after surgery, followed by a gradual recovery in 46 patients. In addition, Ryu et al. [6] reported a decrease in the F_0_, the speaking F_0_, and an increase in jitter and shimmer in 155 patients without RLN damage after thyroidectomy. They reported that these changes were not temporary and might persist for more than 12 months in some cases. A meta-analysis [5] reported a decline in the F_0_, shimmer, and MPT within 3 months, while jitter and NHR remained unchanged after thyroidectomy. Additionally, in the study conducting acoustic analysis on vocal fatigue symptoms [25,26], the F_0_, jitter, and shimmer did not yield consistent results, and they did not prove the correlation between subjective vocal fatigue assessments and acoustic parameters. As such, acoustic analysis through the MDVP does not consistently yield results in the voice evaluation of patients with PTS, suggesting potential limitations.

The cepstrum was first introduced by Bogert et al. [27], and it involves inverse Fourier transforming the log spectrum using a log function on the spectrum. The CPP is the most crucial indicator in cepstral analysis, serving to evaluate the periodicity of the voice [16,28]. Time-based acoustic measurements obtained through the MDVP depend on the precise assessment of voice periodicity. The MDVP is heavily influenced by the F_0_, rendering irregular and aperiodic voice analysis less reliable [29,30,31]. On the other hand, because cepstral analysis can be evaluated without the need for the quantitative measurement of periodicity, it proves useful for analyzing irregular and aperiodic voices. Previous reports have indicated that patients with asthenic voice disorders, such as vocal cord paralysis, exhibit poor periodicity and unclear sound quality, leading to a decrease in CPP values [32,33,34]. In this study, similar to previous research, a decrease in the CPP/a/ and CPP/sen/ was observed without statistical significance. However, the changes in the CPP/a/ demonstrated a significant negative relationship with changes in subjective voice analysis. The L/H ratio is defined as the ratio of frequency spectrum energy below 4000 Hz to frequency spectrum energy above 4000 Hz. In disability speech, the L/H ratio tends to be low [16]; however, the L/H ratio was shown to be high after surgery in our study. It has been found that relying solely on the L/H ratio is insufficient to evaluate the severity of the disorder. Hence, the anticipated utility of the significant negative correlation between the CPP reduction after thyroidectomy and the subjective voice discomfort assessment, along with the CPP/a/, makes it a valuable factor in the voice evaluation of patients with PTS. Further studies, comparing subjective discomfort with cepstrum results in a larger cohort of patients, are deemed necessary for future research.

The EGG involves sending high-frequency current signals through surface electrodes attached to both sides of the thyroid cartilage to measure electrical resistance based on the contact of both vocal cords [35]. The CQ refers to the ratio of vocal cord contact during a cycle. It is reported that, when the vocal cords are strongly closed, as in loud sounds, the closed quotient increases, while in softer sounds or falsetto, it becomes smaller. In this study, there was no significant change in men after surgery, but the CQ value on the EGG was significantly reduced in women. Since female vocal cords are relatively shorter than those of men, they are more susceptible to the effects of weakening surrounding muscles and the swelling of the vocal cords after surgery. This vulnerability is believed to contribute to complaints of discomfort and fatigue in the voice as the contact of the vocal cords decreases.

In the previous literature, a consistent report has been made on the decrease in vocal range attributed to the reduction in peak fundamental frequency after thyroidectomy [5,6]. In addition, a significant correlation was observed between the change in subjective voice disability evaluation and the F_0max_. In this study, for males, the F_0max_ and F_0min_ in the VRP did not show significant changes; however, the F_0max_ significantly decreased in women. Similarly, studies such as Hong et al. [8] showed no change in the F_0min_ after thyroid surgery, but the F_0max_ decreased in 86% of patients. The high-frequency domain is highly affected by voice disorders that occur without nerve damage after thyroidectomy, and the frequency range of the fundamental frequency is higher and wider in women compared to men. Consequently, this is thought to be the reason why women more frequently report subjective voice discomfort after surgery. The cause that can affect the high-frequency region is particularly associated with the EBSLN. Due to variations in the pathways of the EBSLN and the potential for anatomical deformities intersecting with the superior thyroid artery, there is a possibility of damage to the EBSLN during thyroidectomy. Damage to the EBSLN can result in movement disorders of the cricothyroid muscle, leading to voice disorders in the high-pitched range. Although damage to the EBSLN can be diagnosed through the cricothyroid muscle EMG, there is a limitation of our study, in that it is not clinically performed as a routine in patients, making it difficult to determine whether the EBSLN is damaged. However, a decrease in the range, similar to the findings in this study, has been confirmed in previous literature that excluded damage to the EBSLN by performing an EMG [8,36]. The second issue is laryngotracheal fixation. In animal experiments using dogs [37], it was observed that the sternohyoid muscle, and mainly the sternothyroid muscle, pull the laryngeal trachea downward. This action causes a higher amount of air in the subglottic area, increases the pressure of the glottis, shortens the distance between the cricoid and thyroid cartilage, lengthens the vocal cords, and increases the frequency. Conversely, the thyrohyoid muscle played a role in lowering the glottic pressure to shorten the vocal cords and lower the frequency. Therefore, it is believed that direct damage to the subcutaneous muscle, damage due to towing, and postoperative adhesions may occur during thyroidectomy, leading to a decrease in the vocal range.

## 5. Conclusions

The VFI, which reflects the patient’s voice discomfort similar to the VHI, can also capture symptoms of voice fatigue—the main symptom of PTS. A significant correlation was found between the change in subjective voice discomfort after thyroid surgery and the change in the /a/ consonant vocal CPP. Therefore, when evaluating changes in acoustic parameters through the MDVP is not definite, cepstral analysis may serve as useful objective parameters in PTS. To validate the feasibility, further large-group studies would be needed.

## Figures and Tables

**Figure 1 diagnostics-14-00111-f001:**
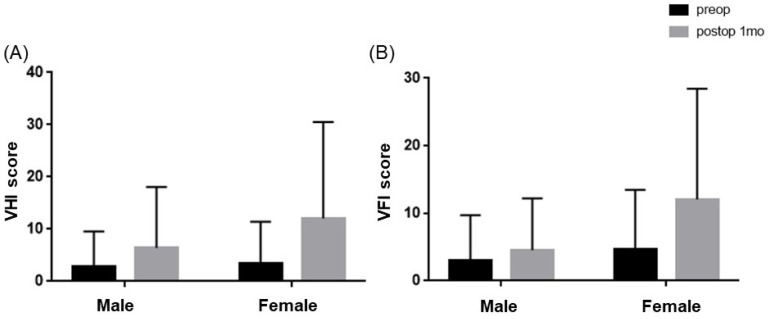
Comparison of subjective voice discomfort (K-VHI and K-VFI) before and after thyroidectomy by gender: (**A**) K-VHI differences between pre- and post-thyroidectomy were significant in females (*p =* 0.001), whereas males did not show any differences (*p =* 0.052). (**B**) K-VFI scores showed significant differences in females after thyroidectomy (*p <* 0.001) rather than in males (*p* = 0.125).

**Table 1 diagnostics-14-00111-t001:** Demographic data of patients who underwent thyroidectomy without recurrent laryngeal nerve injury, categorized by gender.

Characteristics	Total Patients (*n* = 96)	Male Patients (*n* = 37)	Female Patients (*n* = 59)	*p*-Value
Mean age (years)	48.39 ± 10.39	50.6 ± 10.9	47.0 ± 9.9	0.095
Thyroid lobectomy	57/96 (59.4%)	24/37 (64.9%)	33/59 (55.9%)	0.386
Total thyroidectomy	39/96 (40.6%)	13/37 (35.1%)	26/59 (44.1%)	0.386
Central neck dissection	96/96 (100%)	37/37 (100%)	59/59 (100%)	-
Lateral neck dissection	7/96 (7.3%)	5/37 (13.5%)	2/59 (3.4%)	0.104
Benign	19/96 (19.8%)	9/37 (24.3%)	10/59 (16.9%)	0.377
Malignant	77/96 (80.2%)	28/37 (75.7%)	49/59 (83.1%)	0.377
Extrathyroidal extension	35/96 (36.5%)	14/37 (37.8%)	21/59 (35.6%)	0.824
Lymphatic invasion	17/96 (17.7%)	6/37 (16.2%)	11/59 (18.6%)	0.762
Vascular invasion	2/96 (2.1%)	1/37 (2.7%)	1/59 (1.7%)	1.000

**Table 2 diagnostics-14-00111-t002:** Comparison of the auditory–perceptual analysis before and after thyroidectomy based on gender.

	Total Patients (*n* = 96)			Male Patients (*n* = 37)			Female Patients (*n* = 59)		
	Preop	Postop	*p*-Value	Preop	Postop	*p*-Value	Preop	Postop	*p*-Value
G	0.18 ± 0.34	0.14 ± 0.4	0.461	0.19 ± 0.4	0.07 ± 0.2	0.037	0.17 ± 0.3	0.19 ± 0.4	0.813
R	0.30 ± 0.4	0.24 ± 0.6	0.491	0.30 ± 0.4	0.18 ± 0.3	0.203	0.30 ± 0.4	0.29 ± 0.7	0.938
B	0.19 ± 0.4	0.18 ± 0.4	0.921	0.19 ± 0.4	0.15 ± 0.4	0.608	0.19 ± 0.4	0.20 ± 0.4	0.776
A	0.02 ± 0.1	0.06 ± 0.2	0.059	0.03 ± 0.2	0.03 ± 0.2	-	0.08 ± 0.1	0.08 ± 0.3	0.059
S	0.07 ± 0.3	0.04 ± 0.2	0.181	0.07 ± 0.3	0.05 ± 0.2	0.324	0.07 ± 0.2	0.03 ± 0.1	0.255

**Table 3 diagnostics-14-00111-t003:** Comparison of acoustic analysis by the MDVP, cepstral analysis by the ADSV, and vocal range profile (VRP), aerodynamic, and glottal vibration analysis before and after thyroidectomy by gender.

		Total Patients (*n* = 96)			Male Patients (*n* = 37)			Female Patients (*n* = 59)		
		Preop	Postop	*p*-Value	Preop	Postop	*p*-Value	Preop	Postop	*p*-Value
MDVP	F_0_ (Hz)	173.10 ±44.8	174.20 ±48.8	0.768	125.55 ±20.9	125.60 ±28.5	0.990	202.92 ±25.9	204.69 ±30.6	0.630
	Jitter (%)	0.69 ±0.47	9.71 ±88.3	0.320	0.67 ±0.6	23.98 ±142.3	0.326	0.71 ±0.4	0.76 ±0.5	0.552
	Shimmer (%)	2.82 ±1.3	2.29 ±0.9	<0.001	2.65 ±1.1	2.04 ±0.7	0.002	2.92 ±1.4	2.44 ±1.0	0.024
	NHR	0.13 ±0.1	0.14 ±0.1	0.893	0.16 ±0.2	0.15 ±0.1	0.755	0.12 ±0.02	0.13 ±0.1	0.126
	SPI	17.92 ±12.7	20.04 ±12.9	0.147	21.06 ±13.3	24.90 ±15.2	0.153	15.94 ±12.0	17.00 ±10.2	0.538
ADSV	CPP/a/ (dB)	13.20 ±2.3	12.96 ±2.1	0.316	14.65 ±2.8	14.58 ±1.8	0.903	12.38 ±1.4	12.03 ±1.6	0.124
	CPP/sen/ (dB)	8.21 ±1.3	8.35 ±1.3	0.404	9.04 ±1.5	9.51 ±1.9	0.198	7.73 ±1.0	7.69 ±1.3	0.851
	L/H ratio /a/ (dB)	35.79 ±6.0	36.65 ±5.8	0.159	37.45 ±6.0	38.62 ±5.3	0.273	34.84 ±5.9	35.52 ±5.9	0.365
	L/H ratio /sen/ (dB)	173.10 ±44.8	174.20 ±48.8	0.768	125.55 ±20.9	125.60 ±28.5	0.990	202.92 ±25.9	204.69 ±30.6	0.630
VRP	F_0min_ (Hz)	0.69 ±0.47	9.71 ±88.3	0.320	0.67 ±0.6	23.98 ±142.3	0.326	0.71 ±0.4	0.76 ±0.5	0.552
	F_0max_ (Hz)	2.82 ±1.3	2.29 ±0.9	<0.001	2.65 ±1.1	2.04 ±0.7	0.002	2.92 ±1.4	2.44 ±1.0	0.024
Aerodynamic	MPT (s)	0.13 ±0.1	0.14 ±0.1	0.893	0.16 ±0.2	0.15 ±0.1	0.755	0.12 ±0.02	0.13 ±0.1	0.126
EGG	CQ (%)	17.92 ±12.7	20.04 ±12.9	0.147	21.06 ±13.3	24.90 ±15.2	0.153	15.94 ±12.0	17.00 ±10.2	0.538

MDVP, Multi-Dimensional Voice Program; NHR, noise-to-harmonic ratio; SPI, Soft Phonation Index; ADSV, Analysis of Dysphonia in Speech and Voice; VRP, vocal range profile; MPT, maximal phonation time; EGG, Electroglottography; CQ, closed quotient.

**Table 4 diagnostics-14-00111-t004:** Correlation Analysis: Examining changes in VHI (Voice Handicap Index) and VFI (Voice Function Index) by factor before and after thyroidectomy.

	ΔVFI (Factor 1)	*p*-Value	ΔVFI (Factor 2)	*p*-Value	ΔVFI (Factor 3)	*p*-Value	ΔVFI (Total)	*p*-Value
ΔVHI	0.917 **	<0.001	0.622 **	<0.001	0.470 **	<0.001	0.887 **	<0.001

The *p*-value was calculated by Pearson correlation analysis. ** Correlation is significant at the 0.01 level (2-tailed).

**Table 5 diagnostics-14-00111-t005:** Correlation analysis: examining changes in Voice Handicap Index (ΔVHI) and Voice Function Index (ΔVFI) with acoustic parameters before and after thyroidectomy.

	ΔVHI	*p*-Value	ΔVFI	*p*-Value
ΔF_0_ (Hz)	0.002	0.986	0.001	0.995
ΔJitter (%)	0.042	0.687	0.040	0.702
ΔShimmer (%)	−0.221 *	0.030	−0.155	0.132
ΔNHR	−0.005	0.959	0.005	0.961
ΔSPI	−0.108	0.295	−0.112	0.275
ΔCPP/a/ (dB)	−0.315 **	0.003	−0.325 **	0.002
ΔCPP/sen/ (dB)	0.097	0.366	0.075	0.487
ΔL/H ratio /a/ (dB)	0.057	0.600	0.103	0.339
ΔL/H ratio /sen/ (dB)	0.056	0.602	0.045	0.675
ΔF_0min_ (Hz)	−0.008	0.940	−0.022	0.833
ΔF_0max_ (Hz)	−0.393 **	<0.001	−0.452 **	<0.001
ΔMPT (s)	−0.020	0.844	−0.032	0.756
ΔCQ (%)	−0.520 **	<0.001	−0.418 **	<0.001

NHR, noise-to-harmonic ratio; SPI, Soft Phonation Index; CPP, cepstral peak prominence; L/H ratio, low-to-high spectral ratio; MPT, maximal phonation time; CQ, closed quotient. ** Correlation is significant at the 0.01 level (2-tailed); * Correlation is significant at the 0.05 level (2-tailed).

## Data Availability

The data presented in this study are available on request from the corresponding author. The data are not publicly available due to data protection regulations of our institute.

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
