# Peer review of "Early Assessment of Voice Problems in Post-Thyroidectomy Syndrome Using Cepstral Analysis"

_diagnostics, 2024, doi:10.3390/diagnostics14010111_

Round 1

Reviewer 1 Report

Comments and Suggestions for Authors

In this study, the authors aimed to evaluate the utility of conventional subjective assessments such as the VHI and VFI. Additionally, they conducted various objective tests, including MDVP and Cepstral analysis, to assess negative indicators that accurately reflect subjective discomfort in patients' voices.

In particular, the present study highlights the potential of cepstral analysis as an index reflecting subjective voice discomfort, suggesting its promise for a comprehensive PTS voice evaluation.

The study covers some issues that have been overlooked in other similar topics. The structure of the manuscript appears adequate and well divided in the sections. Moreover, the study is easy to follow, but few issues should be improved. Some of the comments that would improve the overall quality of the study are:

a. Authors must pay attention to the technical terms acronyms they used in the text.

b. Conclusion Section: This paragraph required a general revision to eliminate redundant sentences and to add some "take-home message".

Author Response

  1. Authors must pay attention to the technical terms acronyms they used in the text.
    Response 1:  Thank you for your valuable feedback. We appreciate your guidance on the use of technical terms and acronyms in the text. We will carefully review and ensure that they are appropriately addressed in the revised version of the manuscript
  2. Conclusion Section: This paragraph required a general revision to eliminate redundant sentences and to add some "take-home message".
    Response 2:  Thanks for your thoughtful review. We acknowledge your suggestion regarding the conclusion section. Your comments have been immensely helpful, and we will promptly revise the paragraph to eliminate redundancies, such as summarizing the results, and enhance the overall clarity of the ‘take-home message’. We appreciate your valuable input and look forward to presenting an improved conclusion in the reviewed manuscript.

Reviewer 2 Report

Comments and Suggestions for Authors

The present article deals with a problem that becomes particularly important considering the long survival of patients with thyroid neoplasm. Post-thyroidectomy syndrome (PTS) is becoming nowadays a public health problem. For this reason, the present study is of great importance. The design is well described, the material and methods are well argued, the objective and subjective outcomes are well defined and the statistics are adequate. The results are described in the context of details from the current literature and the discussions are a synthesis of recent data about PTS. Reference list is heavily updated. However, in order to increase the impact of the paper I would suggest to discuss the correlations between VHIO, VFI, acoustic, aerodynamic, and glottal vibration analyses and type of thyroid surgery performed.

Author Response

In order to increase the impact of the paper I would suggest to discuss the correlations between VHIO, VFI, acoustic, aerodynamic, and glottal vibration analyses and type of thyroid surgery performed.
Response 1: Thanks for your thoughtful review. As you suggested, there have been numerous concerns regarding the relationship between the type of thyroidectomy and voice changes. In a previous study, Kim et al (1) examined 2,297 thyroidectomy patients without vocal cord paralysis after surgery. They assessed the patients’ voice problems ‘ using the Voice Handicap Index (VHI), and acoustic parameters through MDVP. Additionally, subjective voice problems were evaluated using the Thyroidectomy-Related Voice Questionnaire (TVQ), which addresses questions about voice quality.
In their findings, the type of thyroidectomy, including total and thyroid lobectomy, and the extent of neck dissection did not significantly impact the voice results. In our study, we faced limitations due to a relatively small sample size when analyzing the extent of neck dissection. When focusing on the type of thyroidectomy, our results similarly did not reveal a statistical significant difference (p = 0.248), which did not include in this article.

Ref (1): KIM, Sang-Yeon, et al. Voice change after thyroidectomy without vocal cord paralysis: analysis of 2,297 thyroidectomy patients. Surgery, 2020, 168.6: 1086-1094.